# Associations of Genetic Variants of Methylenetetrahydrofolate Reductase and Serum Folate Levels with Metabolic Parameters in Patients with Schizophrenia

**DOI:** 10.3390/ijerph182111333

**Published:** 2021-10-28

**Authors:** Chun-Hsin Chen, Po-Yu Chen, Cynthia Yi-An Chen, Chih-Chiang Chiu, Mong-Liang Lu, Ming-Chyi Huang, Yen-Kuang Lin, Yi-Hua Chen

**Affiliations:** 1Department of Psychiatry, School of Medicine, College of Medicine, Taipei Medical University, Taipei 110, Taiwan; Eric.ccchiu@gmail.com (C.-C.C.); mongliang@hotmail.com (M.-L.L.); mingchyihuang@gmail.com (M.-C.H.); 2Department of Psychiatry, Wan Fang Hospital, Taipei Medical University, Taipei 116, Taiwan; cyn.placido@gmail.com; 3Department of Psychiatry, Taipei City Psychiatric Center, Taipei City Hospital, Taipei 110, Taiwan; boyu.chen@gmail.com; 4Graduate Institute of Medical Science, School of Medicine, Taipei Medical University, Taipei 110, Taiwan; 5Biostatistics Center, Taipei Medical University, Taipei 110, Taiwan; robbinlin@tmu.edu.tw; 6School of Public Health, College of Public Health, Taipei Medical University, Taipei 110, Taiwan

**Keywords:** folate, methylenetetrahydrofolate reductase, metabolic parameters, one-carbon metabolism, schizophrenia

## Abstract

The one-carbon metabolism pathway is a suitable candidate for studying the genetic and epigenetic factors contributing to metabolic abnormalities in patients with schizophrenia. We recruited 232 patients with schizophrenia and analyzed their serum folate, vitamin B12, and homocysteine levels and metabolic parameters to investigate the associations of genetic variants of methylenetetrahydrofolate reductase (*MTHFR*) and folate levels with metabolic parameters. *MTHFR* C677T and *MTHFR* A1298C were genotyped. Results showed that *MTHFR* 677T allele carriers had lower levels of total cholesterol and low-density lipoprotein cholesterol than those with the 677CC genotype. Metabolic parameters did not differ between *MTHFR* 1298C and 1298AA carriers. Patients with a low folate level had a lower high-density lipoprotein cholesterol level than those with a normal folate level, but the effect disappeared after adjustment for age, sex, and types of antipsychotics used. We found significant interactions between *MTHFR* A1298C and the folate level status (low vs. normal) in terms of body mass index and waist circumference. In conclusion, genetic variants in one-carbon metabolism might play a role in antipsychotic-induced metabolic abnormalities. Prospective studies on drug-naïve, first-episode patients with schizophrenia are warranted to identify key regions of DNA methylation changes accounting for antipsychotic-induced metabolic abnormalities.

## 1. Introduction

Metabolic abnormalities are a serious concern in patients with schizophrenia. Clozapine and olanzapine are antipsychotics that have the highest propensity to induce body weight gain and metabolic abnormalities [1]; however, not all patients treated with clozapine or olanzapine experience metabolic adverse effects [2]. Furthermore, some evidence has shown that increased visceral fat and impaired fasting glucose tolerance are already present in drug-naïve patients with schizophrenia [3,4]. These observations suggest that, in addition to antipsychotic treatment, genetic or epigenetic factors may contribute to metabolic abnormalities in patients with schizophrenia [5,6].

Enzyme abnormalities in one-carbon metabolism have been found in the blood cells of patients with schizophrenia [7,8]. In the one-carbon metabolism pathway, 5,10-methelenetetrahydrofolate reductase (*MTHFR*), a regulatory enzyme, converts 5,10-methylene tetrahydrofolate (5,10-*MTHF*) to 5-methyltetrahydrofolate (5-*MTHF*), which is a key methyl donor for homocysteine remethylation to methionine. *MTHFR* C677T and *MTHFR* A1298C are two common functional genetic variants of *MTHFR* related to *MTHFR* enzyme activity [9,10]. For example, each copy of *MTHFR* C677T T allele reduces *MTHFR* activity by approximately 35% [9]. The low activity of *MTHFR* impedes the conversion of homocysteine to methionine that is used to methylate DNA. In addition, decreased availability to folate may induce homocysteine accumulation because of the inability of homocysteine to be metabolized to methionine. Homocysteine elevation and methionine reduction following low activity of *MTHFR* and low folate availability result in DNA hypomethylation [11,12], which has been shown to be associated with hyperglycemia and low high-density lipoprotein cholesterol (HDL-C) levels [13]. A meta-analysis showed that *MTHFR* 677T carriers, particularly those with low folate levels, are at increased risk for coronary heart disease [14]. Whether the genetic effect specifically related to low folate level holds true for schizophrenia is unknown.

*MTHFR* is one of the susceptibility genes in schizophrenia and type 2 diabetes mellitus [15]. In patients with schizophrenia, the associations between genetic variants of *MTHFR* and metabolic parameters are inconsistent in the literature. Ellingrod et al. found that *MTHFR* 677T carriers were more vulnerable to metabolic syndrome than those with *MTHFR* 677CC in a sample mainly comprising Caucasian patients with schizophrenia [16]. Contrarily, Roffeei et al. found that the *MTHFR* 677T allele was protective against metabolic syndrome in Malaysian and Chinese patients with schizophrenia [17]. Furthermore, the genetic heterogeneity of *MTHFR* among different regions of the world has been demonstrated [18], and the effects of genetic variants of *MTHFR* on homocysteine levels differ among various regions [19]. These findings indicate that ethnic factors should be considered when estimating the associations between genetic variants of *MTHFR* and metabolic syndrome.

Given the evidence that *MTHFR* genotypes potentially affect the metabolic profiles in patients with schizophrenia treated with antipsychotics, and that the effects may be related to low folate levels and may differ among different ethnicities, we explored the associations of *MTHFR* genetic variants and serum folate level with metabolic parameters in patients with schizophrenia treated with antipsychotics in the Han Chinese population in Taiwan. Because low function of *MTHFR* and low folate availability result in DNA hypomethylation, which might contribute to metabolic abnormalities [13], we hypothesized that patients with the minor allele of *MTHFR* (i.e., *MTHFR* 677T or *MTHFR* 1298C) and those with a low folate level may have worse metabolic parameters.

## 2. Material and Methods

### 2.1. Participants and Settings

The study was a part of a screening phase of a randomized trial investigating folate supplementation in patients with schizophrenia (ClinicalTrial.gov identifier: NCT02916121). The trial was conducted in outpatient clinics of Taipei Medical University-Wan Fang Hospital and Taipei City Psychiatric Center from October 2014 to December 2017, in accordance with the Declaration of Helsinki. The protocol was approved by the Research Ethical Committees of both institutes. The inclusion criteria were as follows: (1) age between 20 and 65 years; (2) fulfillment of the DSM-IV-TR criteria for schizophrenia or schizoaffective disorder, as verified by two board-certified psychiatrists; (3) treatment with an antipsychotic agent for at least 6 months or at a stable dose for at least 3 months. The exclusion criteria were as follows: (1) medically unstable status; (2) current vitamin supplementation; (3) current pregnancy or lactation; (4) presence of substance use disorder (including abuse and dependence) based on a clinical interview and urine tests during the preceding year. After initial assessments, a comprehensive description of the study was provided to the eligible participants, and written informed consent for participation was obtained before including them in the study. If patients could not understand the study protocol owing to impaired cognition, we explained the protocol to their caregivers and obtained informed consent under their supervision. The participants could decide to withdraw their informed consents anytime during the study period. Data including age, sex, diagnosis, and age at onset of schizophrenia of the recruited patients were collected from clinical interviews and medical records.

### 2.2. Clinical Assessment

Trained research assistants, who had completed at least 6 h of training for conducting human research or clinical trials each year before recruiting patients, or psychiatrists conducted all the assessments, including height, body weight, sitting blood pressure (BP), and waist circumference.

### 2.3. Laboratory Assays

We withdrew fasting venous blood between 8:00 and 10:00 a.m. Baseline biochemical assays were performed in advance to ensure no abnormality in general biochemical screening. Serum levels of folate, vitamin B12, and homocysteine and metabolic parameters, including fasting plasma glucose (FPG), triglyceride, total cholesterol, low-density lipoprotein cholesterol (LDL-C), and HDL-C, were measured using an automated system. We used a serum folate level of 6 ng/mL as the cutoff to categorize patients into the low folate level (≤6 ng/mL) and normal folate level (>6 ng/mL) [20]. Metabolic syndrome was defined based on the International Diabetes Federation Task Force Criteria; three or more of the following five criteria were required: (1) waist circumference greater than 90 cm in men and greater than 80 cm in women; (2) fasting serum triglyceride levels of 150 mg/dL or above; (3) fasting serum HDL-C levels less than 40 mg/dL in men or less than 50 mg/dL in women; (4) systolic BP higher or equal to 130 mm Hg, diastolic BP higher or equal to 85 mm Hg, or current treatment with antihypertensive medication; (5) an FPG level of 100 mg/dL or above, or current treatment with antihyperglycemic medication [21].

### 2.4. DNA Assays

Genomic DNA was isolated from blood using standard protocols and was stored in a −80 °C refrigerator until genetic assays were conducted. Genetic variants of *MTHFR* C677T (rs1801133) and *MTHFR* A1298C (rs1801131) were genotyped using the ABI StepONE Plus^TM^ Software v2.1 genotyping assay (Thermo Fisher Scientific, Waltham, MA, USA).

### 2.5. Statistical Analysis

We used descriptive statistics to present demographic characteristics and laboratory data, including metabolic parameters, in recruited patients. Independent *t*-test and chi-square test were used to evaluate the differences in continuous variables and categorical variables between groups, respectively. General linear model was used to control for potential confounders to evaluate the main effects of *MTHFR* polymorphisms and serum folate level status, as well as their interaction effects, on metabolic parameters. A *p*-value of <0.05 indicated statistical significance.

## 3. Results

A total of 232 participants were enrolled in our study. Table 1 presents the demographic and clinical characteristics, genetic variants of *MTHFR* C677T and *MTHFR* A1298C, and laboratory data of the participants. Clozapine (35.8%) and olanzapine (13.4%) accounted for the most commonly used antipsychotics. The distributions of genetic variants of *MTHFR* C677T and *MTHFR* A1298C did not violate the Hardy–Weinberger equilibrium.

Table 2 presents the demographic characteristics and laboratory data of patients with different genotypes of *MTHFR* C677T and *MTHFR* A1298C. Patients who were *MTHFR* 677T carriers (CT/TT) had significantly lower levels of total cholesterol, LDL-C, folate, and vitamin B12 and a higher level of homocysteine than those with the CC genotype. Regarding *MTHFR* A1298C, the metabolic parameters were comparable between patients with the 1298C genotype and those with the 1298AA genotype.

When we categorized our patients on the basis of a cutoff level of 6 ng/mL, mean age was younger and there were fewer female patients with low folate levels than those with normal folate levels (shown in Table 3). The percentage of patients using clozapine or olanzapine was higher in patients with low serum folate levels than those with normal folate levels. In addition, patients with a low folate level had significantly lower levels of serum HDL-C and vitamin B12 and higher levels of homocysteine compared with those with a normal folate level. When we further adjusted the effect of age, sex, and type of antipsychotics used, the difference in HDL-C level between the two groups disappeared, whereas the significant difference in the levels of vitamin B12 and homocysteine remained unchanged.

When we further explored the interaction effect between genetic variants of *MTHFR* and folate level status (low vs. normal) on metabolic parameters after adjusting for the effect of age, sex, duration of illness, and type of antipsychotics used, we found significant interaction effects between *MTHFR* A1298C and folate level status on body mass index (BMI) (*p* = 0.01) and waist circumference (*p* = 0.01), but not on other metabolic parameters. Specifically, patients with the *MTHFR* 1298C allele and low folate levels had significantly higher BMI and waist circumference (shown in Table 4). Regarding *MTHFR* C677T, the genotypes did not interact with the folate level status to influence metabolic parameters.

No significant differences were noted in the distributions of genetic variants of *MTHFR* C677T and *MTHFR* A1298C, the folate level status, levels of folate, homocysteine, and vitamin B12, and type of antipsychotics used between patients with metabolic syndrome and those without metabolic syndrome (data not shown).

## 4. Discussion

To the best of our knowledge, this is the first study to investigate the effect of genetic variants of *MTHFR* and serum folate levels, as well as their interactions on metabolic parameters in patients with schizophrenia. Other than the finding that the *MTHFR* 677T allele was associated with lower levels of total cholesterol and LDL-C, we did not detect significant effects of *MTHFR* genotypes or the folate level status on metabolic parameters or metabolic syndrome in patients with schizophrenia. However, a significant interaction effect of *MTHFR* A1298C and the folate level status on BMI and waist circumference was noted.

Contrary to our hypothesis, patients with the *MTHFR* 677T allele had favorable metabolic parameters with significantly lower levels of total cholesterol and LDL-C in our study. In a similar attempt to examine the association between the genetic variants of *MTHFR* and metabolic parameters in patients with schizophrenia, two previous cross-sectional studies on Caucasian patients failed to find any relationship between *MTHFR* C677T and metabolic parameters [16,22]. Ethnicity may be one of the reasons accounting for the differences between studies. Regarding the genetic effects of *MTHFR* polymorphisms on the vulnerability to metabolic syndrome or insulin resistance, existing research has reported inconclusive findings. *MTHFR* 677T carriers in Ellingrod’s study [16] and *MTHFR* 1298C carriers in van Winkel’s study [22] were reported to have higher risks of metabolic syndrome. Additionally, carriers with *MTHFR* 677T or *MTHFR* 1298C allele had a higher level of insulin resistance [23]. By contrast, the *MTHFR* 677T allele seemed to be protective against metabolic syndrome in Roffeei’s study on Asian patients [17]. In our study, no relationship was found between *MTHFR* polymorphisms, whether *MTHFR* C677T or *MTHFR* A1298C, and metabolic syndrome. However, Srisawat et al. found that patients with *MTHFR* 677CC were associated with a greater increase in BMI compared with *MTHFR* 677T carriers in two cohorts of Chinese Han and Spanish Caucasians [24], both of which displayed similar minor allele frequency of *MTHFR* C677T and *MTHFR* A1298C. Taken together, these observations revealed inconclusive results in terms of the role of ethnic factor in the association between genetic variants of *MTHFR* and antipsychotic-induced body weight gain. Therefore, the effects of genetic variants of *MTHFR* on metabolic parameters should be further elucidated.

Notably, we found significant interactions between *MTHFR* A1298C and the folate level status in terms of BMI and waist circumference. Despite the current evidence that folate level is not significantly associated with metabolic syndrome [22] or antipsychotic-induced changes in metabolic parameters [25] in patients with schizophrenia, no study has investigated the possible interaction effect between genetic variants of *MTHFR* and folate levels on metabolic parameters. However, the interaction effect has previously been shown in nonpsychiatric patients. For example, *MTHFR* 677T carriers with a low folate level were associated with a higher risk of low HDL-C in Chinese patients with hypertension [26]. Similarly, a significant higher risk of coronary heart disease was noted in *MTHFR* 677T carriers, especially in those with low folate levels [14]. In addition to metabolic-related indices, the impact from interactions between genetic variants of one-carbon metabolism and folate levels on psychopathology in patients with schizophrenia has been reported [27]. Because the present study is the first to explore the interaction effect between genetic variants of *MTHFR* and folate levels on metabolic parameters in patients with schizophrenia, future studies with a larger sample size are warranted to verify our results.

Medications may affect the associations between genetic variants of *MTHFR* and metabolic parameters. Van Winkel et al. found that the effect of *MTHFR* A1298C on metabolic syndrome was greater in patients treated with clozapine or olanzapine [22]. However, we did not find any significant differences in metabolic parameters between patients with different genetic variants of *MTHFR* (either C677T or A1298C) in subgroups of patients receiving or not receiving clozapine/olanzapine treatment (data not shown).

The results of the associations between the genetic variants of *MTHFR* and metabolic parameters in patients with schizophrenia should be cautiously interpreted because they might be confounded by various factors such as the duration of illness and the medication effect from long-term antipsychotic exposure. Kao et al. found that the *MTHFR* 677C allele was significantly associated with antipsychotic-induced weight gain, but the effect seemed to be prominent in the subpopulation of first-episode patients [28]. Although one cross-sectional study revealed no association between the genetic variants of *MTHFR* and metabolic parameters in first-episode patients with schizophrenia taking short-term antipsychotics (mean, 6 days) [29], two longitudinal follow-up studies found that those with *MTHFR* 677CC gained more body weight or exhibited a more significant increase in BMI compared with *MTHFR* 677T carriers among first-episode patients with schizophrenia following antipsychotic treatment [24,25]. In addition, one recent study investigated the effect of *MTHFR* C677T on antipsychotic-induced changes of body weight gain and BMI, in which *MTHFR* 677C carriers had a higher body weight gain or BMI change than those with *MTHFR* 677TT in Chinese patients with schizophrenia [30]. The effects are different among antipsychotics used, especially in risperidone-treated patients, and more prominent in first-episode patients with schizophrenia [30]. Therefore, *MTHFR* C677T may be significantly associated with antipsychotic-induced body weight changes, especially in the early stage of treatment.

Our study had several limitations. First, we recruited patients who were relatively chronic with a mean duration of illness of nearly 20 years. We did not collect information regarding lifestyle factors, such as diet and physical activities and antipsychotic use, which may confound our results. Second, we did not measure DNA methylation and *S*-adenosyl methionine (SAM), which is the methyl donor of DNA methylation, in the one-carbon cycle. Therefore, whether differences in DNA methylation and SAM exist between patients with different genetic variants of *MTHFR* or between patients with low folate levels and those with normal folate levels is unknown. Previous evidence has shown that *MTHFR* 677TT had the lowest DNA methylation compared with other genetic variants of *MTHFR* C677T in female patients with schizophrenia [31]. Global methylation increased, especially in patients treated with clozapine and olanzapine, and endothelial dysfunction improved after folate supplementation in schizophrenic patients with metabolic syndrome [32]. Certain regions of DNA methylation at birth were associated with obesity and insulin resistance in childhood [33]. In addition, site-specific DNA methylation has been shown to be associated with metabolic alternations in patients treated with antipsychotics [34,35]. Therefore, analysis of DNA methylation is essential when studying the associations between one-carbon metabolism and metabolic abnormalities. Third, the power to detect the effects of genetic variants of *MTHFR* on metabolic parameters was inadequate due to the small sample size. Fourth, we did not recruit healthy participants. Therefore, we could not determine whether there were any differences in the effects of *MTHFR* genetic variants on metabolic parameters between the general population and patients with schizophrenia.

## 5. Conclusions

Contrary to our hypothesis that patients with the minor allele of *MTHFR* (i.e., *MTHFR* 677T or *MTHFR* 1298C) and those with a low folate level may have worse metabolic parameters, we found that the *MTHFR* 677T allele was associated with lower levels of total cholesterol and LDL-C. No other significant effects of *MTHFR* genotypes or the folate level status on metabolic parameters or metabolic syndrome in patients with schizophrenia were noted. However, we found an interaction effect between the genetic variant and folate level on BMI and waist circumference. Genetic variants of one-carbon metabolism might play a role in antipsychotic-induced metabolic abnormalities, especially considering the effect of folate levels. Because of limitations in recruiting chronic patients, lack of a healthy control, and no DNA methylation data in the current study, prospective studies recruiting drug-naïve, first-episode patients with schizophrenia and healthy participants are needed in the future to identify changes in key regions of DNA methylation accounting for antipsychotic-induced metabolic abnormalities.

## Figures and Tables

**Table 1 ijerph-18-11333-t001:** Demographic characteristics, *MTHFR* genotypes, and metabolic parameters of patients with schizophrenia (*n* = 232).

Variables	
Age ^a^ (years), mean ± SD	44.3 ± 10.7
Sex, male/female, *n* (%)	115/117 (49.6/50.4)
Onset age ^b^ (years), mean ± SD	24.1 ± 8.9
Duration of illness ^c^ (years), mean ± SD	19.8 ± 10.5
Antipsychotic use, *n* (%)	
Clozapine	83 (35.8)
Olanzapine	31 (13.4)
Risperidone	25 (10.8)
Haloperidol	22 (9.5)
Aripiprazole	14 (6.0)
Others	57 (24.5)
*MTHFR* C677T (rs1801133)	
C/C, *n* (%)	116 (50.0)
C/T, *n* (%)	97 (41.8)
T/T, *n* (%)	19 (8.2)
*MTHFR* A1298C (rs1801131)	
A/A, *n* (%)	152 (65.5)
A/C, *n* (%)	66 (28.4)
C/C, *n* (%)	14 (6.0)
BMI ^d^ (kg/m^2^), mean ± SD	26.7 ± 5.1
Antipsychotic use, olanzapine/clozapine, *n* (%)	116 (50)
WC ^d^ (cm), mean ± SD	91.0 ± 12.5
SBP ^d^ (mmHg), mean ± SD	124.2 ± 18.3
DBP ^d^ (mmHg), mean ± SD	77.3 ± 12.0
TG (mg/dL), mean ± SD	148.1 ± 87.0
Total cholesterol (mg/dL), mean ± SD	189.6 ± 38.8
HDL-C (mg/dL), mean ± SD	51.3 ± 16.7
LDL-C (mg/dL), mean ± SD	119.6 ± 35.6
FPG (mg/dL), mean ± SD	108.5 ± 39.3
Folate (ng/mL), mean ± SD	8.6 ± 3.9
Homocysteine (μmol/mL), mean ± SD	16.4 ± 14.3
Vitamin B12 (pg/mL), mean ± SD	432.1 ± 266.6

^a^ Two missing data; ^b^ 19 missing data; ^c^ 20 missing data; ^d^ six missing data in these variables. Abbreviations: BMI, body mass index; DBP, diastolic blood pressure; FPG, fasting plasma glucose; HDL-C, high-density lipoprotein cholesterol; LDL, low-density lipoprotein cholesterol; MTHFR, methylenetetrahydrofolate reductase; SBP, systolic blood pressure; TG, triglyceride; WC, waist circumference.

**Table 2 ijerph-18-11333-t002:** Demographic characteristics and metabolic parameters between different genotypes of *MTHFR*.

	*MTHFR* C677T			*MTHFR* A1298C		
	C/C	C/T + T/T	*p*	A/A	A/C + C/C	*p*
	*n* = 116	*n* = 116		*n* = 152	*n* = 80	
Age ^a^ (years), mean ± SD	44.3 ± 10.0	44.4 ± 11.4	0.92	44.5 ± 10.5	43.9 ± 11.0	0.61
Sex, male/female, *n* (%)	54/62 (46.6/53.4)	61/55 (52.6/47.4)	0.36	74/78 (48.7/51.3)	41/39 (51.2/48.8)	0.71
Onset age ^b^ (years), mean ± SD	23.9 ± 8.7	24.3 ± 9.1	0.84	24.1 ± 8.8	24.1 ± 9.0	0.98
Duration of illness ^c^, years, mean ± SD	20.3 ± 9.9	19.3 ± 11.1	0.51	19.9 ± 10.6	19.7 ± 10.4	0.93
Olanzapine/clozapine, *n* (%)	59 (50.9)	55 (47.4)	0.60	78 (51.3)	36 (45.0)	0.36
BMI ^d^ (kg/m^2^), mean ± SD	27.0 ± 5.1	26.3 ± 5.2	0.35	26.4 ± 4.9	27.1 ± 5.7	0.35
WC ^d^ (cm), mean ± SD	91.3 ± 13.3	90.7 ± 11.7	0.70	90.6 ± 11.7	91.8 ± 13.9	0.48
SBP ^d^ (mmHg), mean ± SD	123.6 ± 18.3	124.8 ± 18.5	0.62	124.8 ± 18.4	123.1 ± 18.3	0.50
DBP ^d^ (mmHg), mean ± SD	77.4 ± 11.8	77.2 ± 12.2	0.92	77.1 ± 12.1	77.6 ± 11.9	0.79
TG (mg/dL), mean ± SD	149.0 ± 85.7	147.1 ± 88.6	0.87	150.7 ± 90.0	143.2 ± 81.3	0.53
Total cholesterol (mg/dL), mean ± SD	194.8 ± 40.8	184.4 ± 36.1	0.04	188.6 ± 35.9	191.5 ± 43.9	0.60
HDL-C (mg/dL), mean ± SD	50.7 ± 16.2	51.9 ± 17.2	0.58	51.5 ± 15.7	50.9 ± 18.6	0.79
LDL-C (mg/dL), mean ± SD	125.2 ± 38.4	114.1 ± 31.9	0.02	118.0 ± 34.0	122.8 ± 38.6	0.33
FPG (mg/dL), mean ± SD	109.0 ± 42.4	108.0 ± 36.1	0.85	110.3 ± 44.9	105.0 ± 25.5	0.32
Folate (ng/mL), mean ± SD	9.5 ± 3.9	7.7 ± 3.7	<0.001	8.5 ± 3.9	8.8 ± 3.9	0.49
Low folate, *n* (%)	19 (16.4)	46 (39.7)	<0.001	42 (27.6)	23 (28.7)	0.86
Homocysteine (μmol/mL), mean ± SD	13.3 ± 5.1	19.4 ± 19.0	0.001	17.8 ± 17.0	13.6 ± 5.4	0.03
Vitamin B12 (pg/mL), mean ± SD	480.5 ± 307.5	383.4 ± 207.9	0.005	422.8 ± 265.3	449.8 ± 269.8	0.47

^a^ Two missing data; ^b^ 19 missing data; ^c^ 20 missing data; ^d^ six missing data in these variables Abbreviations: BMI, body mass index; DBP, diastolic blood pressure; FPG, fasting plasma glucose; HDL-C, high-density lipoprotein cholesterol; LDL, low-density lipoprotein cholesterol; MTHFR, methylenetetrahydrofolate reductase; SBP, systolic blood pressure; TG, triglyceride; WC, waist circumference.

**Table 3 ijerph-18-11333-t003:** Demographics, clinical characteristics, and metabolic parameters of patients with low folate level and those with normal folate level.

	Patients with Low Folate Level (≤6 ng/mL)	Patients with Normal Folate Level (>6 ng/mL)	*p*
	*n* = 65	*n* = 167	
Age ^a^ (years), mean ± SD	41.8 ± 11.1	45.3 ± 10.4	0.03
Sex, male/female, *n* (%)	40/25 (61.5/38.5)	75/92 (44.9/55.1)	0.02
Onset age ^b^ (years), mean ± SD	23.5 ± 8.5	24.3 ± 9.0	0.53
Duration of illness ^c^ (years), mean ± SD	17.9 ± 10.1	20.6 ± 10.6	0.09
Olanzapine/clozapine, *n* (%)	90 (53.9)	24 (36.9)	0.02
BMI ^d^ (kg/m^2^), mean ± SD	27.7 ± 5.7	26.3 ± 4.9	0.054
WC ^d^ (cm), mean ± SD	92.9 ± 11.9	90.2 ± 12.7	0.15
SBP ^d^ (mmHg), mean ± SD	123.7 ± 19.9	124.4 ± 17.8	0.80
DBP ^d^ (mmHg), mean ± SD	78.0 ± 12.4	77.0 ± 11.8	0.57
TG (mg/dL), mean ± SD	149.6 ± 77.8	147.5 ± 90.5	0.87
Total cholesterol (mg/dL), mean ± SD	188.2 ± 41.3	190.1 ± 37.9	0.74
HDL-C (mg/dL), mean ± SD	47.6 ± 12.3	52.7 ± 18.0	0.037
LDL-C (mg/dL), mean ± SD	121.8 ± 36.2	118.8 ± 35.5	0.57
FPG (mg/dL), mean ± SD	107.1 ± 33.1	109.0 ± 41.5	0.73
Homocysteine (μmol/mL), mean ± SD	25.1 ± 23.5	13.0 ± 5.3	<0.001
Vitamin B12 (pg/mL), mean ± SD	351.6 ± 208.2	463.0 ± 280.3	0.004

^a^ Two missing data; ^b^ 19 missing data; ^c^ 20 missing data; ^d^ six missing data in these variables. Abbreviations: BMI, body mass index; DBP, diastolic blood pressure; FPG, fasting plasma glucose; HDL-C, high-density lipoprotein cholesterol; LDL, low-density lipoprotein cholesterol; SBP, systolic blood pressure; TG, triglyceride; WC, waist circumference.

**Table 4 ijerph-18-11333-t004:** Interactions between *MTHFR* A1298C and folate levels in terms of body mass index and waist circumference in patients with schizophrenia.

	*MTHFR* 1298AA	*MTHFR* 1298 AC + CC	*p* ^a^
BMI (kg/m^2^), mean ± SD			0.01
Low folate level	25.9 ± 4.6 (*n* = 38)	30.0 ± 6.7 (*n* = 20)	
Normal folate level	26.4 ± 5.0 (*n* = 97)	26.1 ± 5.0 (*n* = 56)	
WC (cm), mean ± SD			0.01
Low folate level	89.4 ± 10.9 (*n* = 38)	98.7 ± 12.4 (*n* = 20)	
Normal folate level	90.7 ± 12.1 (*n* = 97)	89.3 ± 13.4 (*n* = 56)	

^a^ Interactions between metabolic profiles and status of folate level on body mass index and waist circumference. Abbreviations: BMI, body mass index; MTHFR, methylenetetrahydrofolate reductase; WC, waist circumference.

## Data Availability

The data presented in this study are available on request from the corresponding author.

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
