# Peer review of "Associations of Genetic Variants of Methylenetetrahydrofolate Reductase and Serum Folate Levels with Metabolic Parameters in Patients with Schizophrenia"

_ijerph, 2021, doi:10.3390/ijerph182111333_

Round 1

Reviewer 1 Report

Dear Editor,
I really appreciate the opportunity to review the manuscript ijerph-1416564 entitled:
"Associations of Genetic Variants of Methylenetetrahydrofolate Reductase and Serum Folate Levels with Metabolic Parameters in Patients with Schizophrenia"

I commend the authors for describing this critical and timely issue. The paper is interesting and well written; however, I would like to highlight some issues that merit revision:

1. I have noted some typos in the text, e.g. "Enzyme abnormalities of in one-carbon", and so on; please, correct
2. The paper needs to be re-read carefully by the authors. The use of English is somewhat unclear, and this detracts from the paper's readability
3. The duration of illness presents a large SD in relation to the mean age, (17.9 ± 10.1 and 20.6 ± 10.6), this could be a factor that generates bias as patients with very different duration of disease are lumped together, it would be interesting if the authors added a stratified analysis for 2-3 bands of disease duration (e. 5-10-20 or more years)

Author Response

  1. I have noted some typos in the text, e.g. "Enzyme abnormalities of in one-carbon", and so on; please, correct

RE: Thank you for your careful review. We have checked the whole manuscript and corrected some typos in the manuscript.

  1. The paper needs to be re-read carefully by the authors. The use of English is somewhat unclear, and this detracts from the paper's readability

RE: Thank you for your comment. We have revised the manuscript to improve its readability.

  1. The duration of illness presents a large SD in relation to the mean age, (17.9 ± 10.1 and 20.6 ± 10.6), this could be a factor that generates bias as patients with very different duration of disease are lumped together, it would be interesting if the authors added a stratified analysis for 2-3 bands of disease duration (e. 5-10-20 or more years)

RE: Thank you for your precious comment. Duration of illness may be a confounder to affect the genetic effects of MTHFR on metabolic parameters. We stratified the duration of illness into 3 strata, i.e. 0-9, 10-19, and more than 20, with sample size of 37, 66, and 109, respectively. Because of the smaller sample size in each stratum, no significant effects of genetic variants of MTHFR on metabolic parameters were found. If we stratified the duration of illness into 2 strata, <20 and >= 20 with sample size of 103 and 109, respectively, no significant effects of genetic variants of MTHFR on metabolic parameters were found, too. Because the sample size is not large enough to stratify participants by duration of illness, we keep the data in original form.

Reviewer 2 Report

Dear Authors,

It was a very interesting read, congratulations. However, three minor points to consider that might result in improving the manuscript:

  1. What were the qualifications of the "trained research assistants", who were they? Please elaborate and expand the paragraph (110).
  2. What anthropological measurements were checked (111)? Please explain clearly in the methodology section so the readers know before heading to the results section.
  3. What were the other antipsychotics used by your patients? (table 1) Please expand the table, as n=57 is quite a large group that accounts for almost 1/4 of your patients. This is a very interesting information for the readers based that the study was conducted on Taiwanese population, and the journal is focused on public health.

Best Regards

Reviewer

Author Response

  1. What were the qualifications of the "trained research assistants", who were they? Please elaborate and expand the paragraph (110).

RE: Thank you for your comment. These research assistants should receive at least 6 hours of training for conducting human research or clinical trials each year before recruiting patients. We have added it into the paragraph as the following:

“Trained research assistants, who had completed at least 6 hours of training for conducting human research or clinical trials each year before recruiting patients,…”

  1. What anthropological measurements were checked (111)? Please explain clearly in the methodology section so the readers know before heading to the results section.

RE: Thank you for your comment. We have deleted the term of “anthropological measurement” and added height and body weight in this paragraph.

“…all the assessments, including height, body weight, sitting blood pressure (BP), and waist circumference.

  1. What were the other antipsychotics used by your patients? (table 1) Please expand the table, as n=57 is quite a large group that accounts for almost 1/4 of your patients.

RE: Because there were a lot of antipsychotics patients used in the current study, we listed antipsychotics, which more than 5% patients used, in Table 1. The other antipsychotics included quetiapine (n = 6), sulpiride (n = 11), paliperidone (n = 9), ziprasidone (n = 2), flupentixol (n = 8), zotepine (n =1), amisulpiride (n = 6), clotiapine (n = 2), zuclopenthixol (n = 1), trifluoperazine (n = 1), and chlorpromazine (n = 5). There were 5 patients without antipsychotics information. If we expanded all the antipsychotics in Table 1, the table will become too large. Therefore, we keep the table unchanged.

Round 2

Reviewer 1 Report

The paper is very interesting and well written, methodologically unexceptionable, and the new implementations provide a valid contribution to the work. Every requested correction has been done, and the manuscript is now suitable for publication